# Targeting Metabolic Vulnerabilities in Epstein–Barr Virus-Driven Proliferative Diseases

**DOI:** 10.3390/cancers15133412

**Published:** 2023-06-29

**Authors:** Nicole Yong Ting Leung, Liang Wei Wang

**Affiliations:** Singapore Immunology Network (SIgN), Agency for Science, Technology and Research (A*STAR), 8A Biomedical Grove, Immunos #04-06, Singapore 138648, Singapore

**Keywords:** Epstein–Barr virus, EBV, cancer metabolism, oncometabolic pathways, cancer biology, latency, lytic infection

## Abstract

**Simple Summary:**

Discovered in 1964, Epstein-Barr virus was the first human cancer-causing virus identified. Cancer cells reprogram metabolic pathways to augment their ability to support abnormal rates of proliferation and promote spread through metastatic invasion. Over the course of an infection, EBV drastically alters the metabolic landscape within its host cell, leading to a variety of downstream effects such as contributing to uncontrolled and upregulated cell growth—a hallmark of cancer. In this review, we will provide a comprehensive overview of research hitherto conducted on the means and impact of various metabolic changes that EBV induces, as well as discuss existing and potential treatment options targeting vulnerabilities in EBV-transformed metabolism.

**Abstract:**

The metabolism of cancer cells and Epstein–Barr virus (EBV) infected cells have remarkable similarities. Cancer cells frequently reprogram metabolic pathways to augment their ability to support abnormal rates of proliferation and promote intra-organismal spread through metastatic invasion. On the other hand, EBV is also capable of manipulating host cell metabolism to enable sustained growth and division during latency as well as intra- and inter-individual transmission during lytic replication. It comes as no surprise that EBV, the first oncogenic virus to be described in humans, is a key driver for a significant fraction of human malignancies in the world (~1% of all cancers), both in terms of new diagnoses and attributable deaths each year. Understanding the contributions of metabolic pathways that underpin transformation and virus replication will be important for delineating new therapeutic targets and designing nutritional interventions to reduce disease burden. In this review, we summarise research hitherto conducted on the means and impact of various metabolic changes induced by EBV and discuss existing and potential treatment options targeting metabolic vulnerabilities in EBV-associated diseases.

## 1. Introduction

In less than a year, the discovery of Epstein–Barr Virus (EBV) as the first human tumour virus will commemorate its 60th anniversary [1]. While most commonly known to be a cause of infectious mononucleosis, many studies over the years have identified EBV to be associated with a variety of human diseases that include Burkitt’s Lymphoma (BL), diffuse large B-cell lymphoma (DLBCL), EBV-associated gastric carcinomas (EBVaGC) and nasopharyngeal carcinoma (NPC), Hodgkin Lymphoma, post-transplant lymphoproliferative disease (PTLD) and even autoimmune disorders like multiple sclerosis [2,3,4]. Advances have also been made to develop treatments for EBV-related malignancies, including the use of chemotherapy, immunotherapy and gene therapy [5]. In fact, the first successful chemotherapy drugs—aminopterin, as well as its successor, the commonly used methotrexate—were developed for the treatment of paediatric cancers including EBV-positive BL and acute lymphoblastic lymphoma (ALL) [6,7]; to this day, methotrexate remains a relatively effective therapy against BL and a number of autoimmune conditions, such as rheumatoid arthritis [8].

EBV—also known as human herpesvirus 4—is a double-stranded DNA gammaherpesvirus with a genome of approximately 173 kilobase pairs and has the characteristic ability of herpesviruses of establishing and maintaining lifelong persistency in their hosts through virus-encoded latency programmes [9]. The life cycle of EBV is thought to begin with the primary infection of B lymphocytes, with an intermediate phase involving epithelial cell infections [10]. Less commonly, EBV has also been reported to infect natural killer (NK), T and even smooth muscle cells, although the molecular basis of infection remains poorly understood in these cell types [11]. These infections typically result in virus latency with no overt production of infectious virions. Nonetheless, EBV can be a major driver of host signalling and metabolism during its latent life cycle in various cancers and lymphoproliferative disorders; EBV-encoded gene products can reprogram cellular metabolism to generate substrates that feed into anabolic processes that ultimately fuel growth and proliferation. Like other viruses, EBV can also enter the lytic cycle and has evolved strategies to maximise virion production for spreading to neighbouring cells and between host individuals [12]. To achieve this, EBV alters host cell metabolism to support virus particle assembly over the course of lytic infection involving the coordinated activation of genes in three temporal classes—Immediate Early (IE), Early (E), and Late (L) [13]. Viral lytic genes intersect extensively with metabolic pathways, e.g., viral E gene products mediate viral DNA replication and require nucleotide substrates for the synthesis of new daughter genomes, viral L genes encode most structural components of the virion and therefore require sizeable pools of available amino-acyl tRNAs and robust protein synthesis and quality check machinery.

A key evolutionary impetus for EBV to enter latency is to enable immune evasion and life-long persistence in the host. Latency can be categorised into four main programmes ranging from Latency 0 to III depending on the type of viral genes expressed at the protein level, with Latency III cells expressing the most viral genes [14,15]. In order to establish and maintain the state of latency, the virus modifies the metabolome of the infected cell, enabling continuous growth and proliferation while also avoiding immune detection. Such changes in host cell metabolism occur in both catabolic and anabolic pathways throughout tumorigenesis and the tumour cells’ subsequent growth and survival. Hence, there is great therapeutic promise in targeting particular metabolic factors in EBV-driven tumour cells. In this review, we will survey key metabolic pathways known to be targeted by EBV and provide an overview of clinical trials involving metabolism-regulating drugs against EBV-driven malignant diseases.

## 2. Metabolic Reprogramming in Latency

### 2.1. Molecular Metabolic Mechanisms Underlying Cell Persistence

#### 2.1.1. DNA Replication in Latency

EBV is capable of driving growth and proliferation in infected cells even as they enter into a state of latency with markedly reduced viral gene expression [14]. As these cells proliferate, there is a requirement to replicate both host and viral DNA genomes. In turn, this necessitates the availability of deoxyribonucleotides—the building blocks of genetic material. EBV supports this by upregulating pathways related to nucleotide synthesis and the uptake of nucleotide precursors, thereby augmenting intracellular pools of nucleotides for DNA replication.

In vitro infection of primary B cells is often used as a model of PTLD as the cells exhibit latency III gene expression patterns, as are seen in PTLD patients [16]. Newly infected B cells undergo an early hyperproliferative state which results in a significant reduction in purine deoxyribonucleoside triphosphate (dNTP) pools. As a result, an initial transient state of cell cycle arrest arises from the purine shortfall [17]. Restoration of the purine pools causes these B-cells to enter a state of hyperproliferation [17]. EBV enhances purine production through upregulation of the serine-catabolising mitochondrial one-carbon (1C) pathway, where tetrahydrofolate (THF) acts as a 1C-group carrier in the synthesis of purines and thymidylate. In this pathway, methylene tetrahydrofolate dehydrogenase (MTHFD2) is the rate-limiting enzyme. Both Epstein–Barr virus nuclear antigen 2 (EBNA2), one of six viral-encoded EBV nuclear transcription factors expressed in latency III, and host-encoded MYC have been found to bind to the *MTHFD2* promoter, driving mitochondrial 1C flux as a result [18].

Alongside purine synthesis fuelled by 1C metabolism, EBV-infected B cells require de novo pyrimidine synthesis to support their hyperproliferative state. EBV also employs EBNA2 to drive the upregulation of cytidine triphosphate synthase 1 (CTPS1) through MYC and non-canonical NF-κB signalling [19]. Both CTPS1 and its isozyme CTPS2 use ATP to catalyse the conversion of glutamine and aspartate to uracil monophosphate (UMP)—a precursor for synthesizing CTP de novo [19]. The upregulation of CTPS1 was found to be crucial for the proliferation of EBV-transformed B cells, where a depletion in CTPS1 presented a significant decrease in proliferative ability in GM12878 lymphoblastoid cell lines (LCLs) (Latency III) and Daudi BL lymphocytes (Latency I) [19]. While depletion in CTPS2 alone did not give rise to significant growth inhibition in these cells, dual depletion of both CTPS1 and CTPS2 resulted in near-complete cessation of proliferation with greater significance compared to single CTPS1 knockdown [19]. This underscores the importance of CTPS1 and CTPS2 in cooperating to meet cellular CTP demand, thus sustaining hyperproliferation in EBV-transformed B cells.

CTP can also be taken up via salvage pathways, where import and subsequent phosphorylation of cytidine can be carried out by either uridine-cytidine kinase 1 (UCK1) and UCK2, converting cytidine to cytidine monophosphate (CMP) or uptake of uridine and cytidine deaminase (CDA) conversion of cytidine to uridine to be catalysed by UCK1/2 to UMP [19]. UCK2 was shown to be significantly upregulated post-EBV infection and produce greater levels of UMP. Though UCK has been found to be associated with viral EBNA3A, the exact outcomes and pathways involved in this association remain incompletely understood [20]. More studies need to be undertaken to better understand the role EBNA3A might play in its association with UCK.

#### 2.1.2. Gene Transcription and Protein Translation in Latency

The metabolism of methionine was established to be key in directing EBV epigenetics and latency in EBV-infected B cells. Methionine is an essential amino acid that serves a variety of functions ranging from protein synthesis to the epigenetic control of genes via DNA methylation. Crosstalk between serine and methionine metabolism exists in cancer, where an absence of methionine can augment serine metabolism in order to recycle homocysteine to methionine to supplement the availability of methionine for the methionine cycle [21]. With the high demand for methyl groups in EBV-related B-cell cancers such as BL, the flux of both the methionine and folate cycles have to be consistently maintained. This is evidenced by observations of methionine-restricted conditions resulting in outcomes such as the impairment of B cell transformation arising from EBV infection and the reactivation of lytic and latent membrane protein gene expression in BL cells due to hypomethylation of the EBV genome [22].

Serine is an important donor of one-carbon (1C) to the one-carbon metabolic cycle [23]. The mitochondrial 1C pathway has been found to be a target of EBV, where many of its associated enzymes such as MTHFD2 exhibited upregulation as a result of EBV infection, leading to B-cell transformation. In B-cells freshly infected with EBV, EBNA2—a key regulator of metabolic reprogramming in B cells—activates *MYC* gene expression and MYC goes on to transactivate a number of key genes implicated in serine metabolism, e.g., *SLC1A4* and *SLC1A5* for the import of serine, dihydrofolate reductase (*DHFR*) that converts dihydrofolate to tetrahydrofolate and the molecular carrier responsible for conjugating 1C units [18,23]. These alterations along with sideroflexin 1 (SFXN1) functioning as a mitochondrial serine transporter allow for increased serine consumption to support the demands of cellular hyperproliferation, one of which includes 1C flux where serine is required as a precursor of 1C in the mitochondrial 1C pathway [24].

Though amino acids such as glycine, serine and tryptophan have also been found to be significantly upregulated in GC, the changes in amino acid metabolism of EBVaGC have not yet been well established. A multi-omics study by Yoon et al. has shown downregulation of amino acid biosynthesis genes and upregulation of all amino acids (excluding cysteine) in EBVaGC compared to EBV-negative GC (EBVnGC), indicating a role for the virus in reprogramming GC metabolism [25]. Curiously, EBVaGC is classically considered to be a latency I type tumour; the main viral gene products EBNA1 and non-coding RNAs may be effector molecules targeting metabolic genes. The pathways and mechanisms that regulate these changes have yet to be studied in detail, warranting deeper investigation into these differences to better understand the effect EBV has on GC.

#### 2.1.3. Carbohydrate Metabolism

EBV-positive cancer cells remodel host pathways to drive growth and proliferation. These entail increased demands for carbohydrates, lipids and sterols. While the best-known functions of these molecules are in energy transduction and membrane biogenesis, they can also play important roles in post-translational modifications, e.g., glycosylation, palmitoylation and isoprenylation to confer additional biological activities to proteins. As such, the virus has evolved mechanisms to promote the uptake and biosynthesis of these crucial molecules.

The metabolic shift towards enhanced glycolysis is long known to be a hallmark of cancer. Termed the Warburg effect, cancer cells preferentially utilise aerobic glycolysis, significantly increasing their glucose uptake and lactate production, in spite of the presence of functional mitochondria and sufficient oxygen that permits more efficient energy generation through oxidative phosphorylation [26,27]. Aerobic glycolysis produces important intermediates, e.g., 3-phosphoglycerate, which feeds into oncogenic de novo serine synthesis and can also generate the oncometabolite D-2HG via PHGDH [28]. EBV upregulates aerobic glycolysis in newly infected B cells [18,29,30]. Latent membrane protein 1 (LMP1), a functional and constitutively active homologue of human CD40, is a key promoter of glycolytic flux in EBV-related malignancies [31]. In EBV-associated NPC, LMP1 enhances aerobic glycolysis by downregulating HoxC8 and upregulating glucose transporter 1 (GLUT-1), most significantly in both transcription and translation via mechanistic targeting of the rapamycin complex 1 (mTORC1)/NFκB pathway compared to the other glucose transporter family members [32,33]. LMP1 also enhances the synthesis of hypoxia-induced factor 1 alpha (HIF1α) in LCLs alongside EBNA-LP (also referred to as EBNA5) and EBNA3 binding of prolylhydroxylases 1 and 2 (PHD1 and 2), respectively, to prevent HIF1A hydroxylation and subsequent degradation in these EBV-transformed B-cells [29]. These changes enable the stabilisation of HIF1A, allowing for continued transactivation of genes related to aerobic glycolysis when its otherwise hydroxylated form results in proteasomal degradation [34]. Latent membrane protein 2A (LMP2A) has also been attributed to an increased extracellular acidification rate (ECAR) in LMP2A-positive NPC compared to parental cells, indicative of elevated levels of glycolysis [35]. However, the molecular basis behind this increase has yet to be determined.

Besides LMPs, the presence of several EBV-Encoded BamH I-A Rightward Transcripts (BARTs)—forming the majority of microRNAs (miRNAs) encoded by EBV—have been associated with enhanced glycolytic activity due to their ability to activate the PI3K/AKT pathway via downregulation of Phosphatase and Tensin Homolog deleted on Chromosome 10 (PTEN) (miR-BART4 and miR-BART7-3p) as well as activating MYC (also referred to as c-Myc) (miR-BART7-3p) [36,37,38]. Regardless, direct causal links between these BARTs and the promotion of aerobic glycolysis have not been drawn so far. Contrarily, miR-BART1-5P has clear evidence of its involvement in enhancing glycolysis via inhibition of the α1 catalytic subunit of AMP-activated protein kinase (AMPKα1), leading to activation of downstream mTOR and HIF1a to ultimately increase glycolysis [39]. Despite the current understanding of aerobic glycolysis in NPC, drugs targeting this particular aspect of NPC have yet to be identified [40]. As targeting glycolysis harms not just tumour cells but healthy cells as well, it is an understandably complex undertaking. Nevertheless, should focused targeting against aerobic glycolysis in cancer cells be achieved, there is a strong potential in research towards novel treatment options against EBV-associated NPC.

Due to elevated glucose uptake in cancer tissues, positron emission tomography or computed tomography using 2-deoxy-2-[18F] fluoro-D-glucose (FDG-PET/CT) has been established as a standard to diagnose and observe the progression of a variety of cancers. These range from lung, oesopharyngeal, colorectal and even lymphomas [41]. Neoplasms arising from EBV-related lymphatic infections have shown to be significantly FGD-avid, making the FDG-PET/CT scan a viable technique for monitoring lymphoid development, and by extension, tumour burden in patients [42].

Interference with the tricarboxylic acid (TCA) cycle has also become an increasingly popular approach in handling EBV-positive cancers. The TCA cycle is best known to receive acetyl-CoA and transform it into various anaplerotic intermediates to be used in other anabolic reactions. The availability of glutamine is important for the reaction catalysed by glutaminase, where it converts glutamine to glutamate which can then be used to generate α-ketoglutarate that feeds into the TCA cycle [43]. An avenue EBV utilises for increased extracellular uptake of glutamine is through the amino acid transporter SLC1A5, previously described to be upregulated by EBNA2 [18,44]. Additionally, LMP1 promotes the translocation of GLUT towards the plasma membrane, contributing to increased glutamine uptake in NPC cells [45]. One well-known and clinically administered group of inhibitors acting against cancer TCA metabolism are glutaminase inhibitors [46,47]. It is worth noting that current versions of asparaginase also have significant glutaminase activities, which may account for its unusual efficacy with asparagine synthetase (ASNS)-positive tumours, including those that are EBV-positive, e.g., natural killer/T cell lymphomas, where asparaginase is used as part of the SMILE regimen [48,49].

#### 2.1.4. Lipid and Sterol Metabolism

While normal cells preferentially obtain fatty acids through uptake from extrinsic sources, cancer cells have a proclivity to generate their own fatty acids via de novo lipogenesis. As EBV induces the immortalisation of B cells; fatty acid synthase (FASN) levels and, by extension, lipogenesis become significantly upregulated. LMP1 was implicated in the induction of FASN in this model, as well as in EBV-associated NPC [31]. FASN is downstream of acetyl-CoA carboxylase 1 (ACC1) and has the ability to catalyse palmitate synthesis—a saturated fatty acid important for its roles in ensuring membrane integrity and as a substrate for protein palmitoylation. While mutation of LMP1 at its palmitoylation site does not alter its association with lipid rafts, there is a possibility that palmitate and its downstream products play a crucial role in forming said rafts for LMP1 signalling [50]. Hence, there exists an interdependency between LMP1 upregulation of FASN and its reliance on intracellular lipid rafts for its signalling [51].

Elevated flux through the mevalonate (MVA) and, by extension, the cholesterol biosynthesis pathways has been increasingly associated with being characteristic of cancers. This pathway utilises acetyl-CoA for the production of isoprenoids and sterols that are essential for cancer cell proliferation. MVA is synthesised from hydroxymethylglutaryl coenzyme A (HMG-CoA) and is rate-limited by HMG-CoA reductase (HMGCR) [52]. Notably, HMGCR is the pharmacological target of statins, a widely prescribed class of drugs meant to reduce circulating cholesterol in humans. Farnesyl pyrophosphate (FPP)—that could be used for both squalene synthesis alongside cholesterol biosynthesis—as well as geranylgeranyl pyrophosphate (GGPP) are both derived from MVA, in which the outgrowth of EBV-infected B cells hinges on the availability of GGPP. Rab13, a protein from the family of Ras oncogenes, is highly upregulated by EBV via EBNA3C and is activated by GGPP through geranylgeranylation [53]. This allows for post-transcriptional regulation of the trafficking and signalling of proteins with critical involvement in EBV-mediated B cell transformation as well as latency stabilisation, namely of LMP1 and LMP2A respectively, ultimately supporting Latency III LCL proliferation and survival.

#### 2.1.5. Redox Homeostasis and Senescence

To support high rates of anabolism and redox homeostasis, cancer cells have to establish immense redox capacities, particularly high NADPH/NADP and low NADH/NAD ratios; high levels of NADPH favours reductive synthesis and the accumulation of biomass along with enhanced defence against redox stresses, while increased NAD abundance promotes catabolic flux to generate NADH for energy transduction.

High levels of glucose-6-phosphate dehydrogenase (G6PD) contribute to maintaining NADPH levels via the catalysis of G6P to 6-phosphogluconate. This process enables cells to utilise G6PD-induced NADPH counter toxicity through reduced glutathione, a known antioxidant [54]. In EBV-associated NPC cells, EBV-miR-BART1 significantly upregulates G6PD expression, allowing these tumour cells to better protect against oxidative stress through the antioxidative function of reduced glutathione [55].

While pronounced upregulation of the cholesterol and lipid pathways are crucial for EBV-induced B cell remodelling (see Section 2.1.4), it also generates significant levels of lipid free radicals as by-products. The accumulation of excessive lipid reactive oxygen species (ROS), also referred to as lipid peroxides, in cells triggers ferroptosis—a form of iron-dependent nonapoptotic cell death [56]. The de novo synthesis and reversible oxidation/reduction of glutathione are key in redox defence. Newly-infected B cells and NPC cells are highly reliant on cystine transporter SLC7A11 and glutathione peroxidase 4 (GPX4) to strongly induce glutathione synthesis, preventing deleterious consequences of redox imbalance and reducing cellular sensitivity to ferroptosis [57]. SLC7A11 imports cystine that would be converted to cysteine, a rate-limiting amino acid required for glutathione synthesis, in the reducing environment of the cytoplasm [57]. Newly infected B cells are also capable of synthesising glutathione via the 1C pathway, where 1C-derived glycine would be utilised for the synthesis of glutathione [57]. GPX4 then employs glutathione as the key reducing agent for the resolving of lipid ROS to non-toxic lipid alcohols [58].

Beyond the well-established roles of oxidative phosphorylation (OXPHOS) in generating adenosine triphosphate (ATP) for use as energy, it also has a less appreciated additional role of maintaining the redox balance of NAD/NADH in cells. As B cells transform to become LCLs, OXPHOS and, by extension, oxygen consumption have been observed to be significantly upregulated by EBNA2 and LMP1 [18,30]. On the other hand, LMP1 upregulates the expression of DNA methyltransferases 1 (DNMT1) and promotes the latter’s translocation to the mitochondria in NPC, leading to downregulation in OXPHOS arising from methylation of OXPHOS electron transport chain complex genes, inhibiting protein expression [59]. It is possible that LMP1 regulates the NAD/NADH balance via the upregulation of NAD(P)H oxidase (NOX) in NPC cells instead of via the OXPHOS pathway [60].

The metabolism of 1C compounds also plays a key role in the production of reducing cofactors required in other metabolic pathways (i.e., NADH and NADPH). It involves both the folate and methionine cycles; at its core lies the folate cofactor by which it is mediated. As touched on previously, EBV-infection of primary B cells upregulates a mitochondrial 1C protein not typically expressed in resting B cells and normal adult cells, MTHFD2, via viral EBNA2 and MYC. Knockout of MTHFD2 in EBV-infected LCLs thus demonstrates a significant decrease in cell proliferation as the lack of the enzyme limited the availability of consumable NADPH [18]. The importance of redox homeostasis of NADPH via the upregulation of MTHFD2 in transformed cells marks the potential of the protein serving as a prospective chemotherapeutic target of low toxicity.

#### 2.1.6. Epigenetic Control and Post-Translational Modification of Tumour Suppressor Proteins

Cancer cells require continuous replication, even in the face of aberrant mutations—EBV cancers are not an exception. To combat the effects of tumour suppressor genes (TSGs) that act as obstacles to their proliferation, EBV employs epigenetic silencing in the form of histone methylation reliant on the availability of universal methyl donor SAM [61]. EBV subverts pathways relating to methionine metabolism to enhance the production of SAM, where the flux of the cycle continues to convert S-adenosylhomocysteine (SAH) to homocysteine that can be metabolised to regenerate methionine, thus feeding back into the SAM cycle [62,63].

LMP2A has been shown to be capable of inducing DNMT1 by phosphorylation of STAT3 in EBVaGC cells while LMP1 also demonstrated the ability to upregulate DNMT1 to methylate the promoter of *PTEN* [59,64]. PTEN downregulation induces activation of the PI3K/AKT pathway, culminating in increased glycolytic flux [65]. A study by Saha et. al. found that DNMT3B and DNMT3L were preferentially upregulated in LCLs compared to resting B lymphocytes, with DNMT3L acting as an enhancer of DNMT activity by increasing their binding ability to SAM, the methyl group donor [66,67,68].

EBV in NPC is capable of utilising miR-BART5-3p to expedite the degradation of p53, a tumour suppressor protein known to regulate cell division [69]. Supporting p53 degradation, EBNA1 and 3C have also demonstrated to capacity to destabilise p53 by keeping it ubiquitinated or by directly interfering with its function by binding to p53′s C-terminal DNA binding domain, respectively [70,71]. TSG p53 is known to play important roles in regulating metabolism in cells by impacting a wide variety of metabolic pathways which include glycolysis, OXPHOS and fatty acid synthesis [72]. Hence, EBV miRNA regulation of p53 may have foreseeable indirect effects on cellular metabolism.

### 2.2. Molecular Metabolic Mechanisms Driving Immune Evasion and Metastasis

#### 2.2.1. Mammalian Target of Rapamycin (mTOR) Signalling to Metastatic Regulators

Besides LMP1 activation of the mTORC1/NFκB pathway in NPC, LMP1 is also capable of activating the PI3K/mTOR/AKT pathway, bringing about an epithelial-to-mesenchymal transition (EMT) and contributing to metastasis in EBV-infected NPC [73]. LMP1-activated mTORC1 induces high levels of aldehyde dehydrogenase (ALDH), a known cancer stem cell biomarker where high levels are associated with poorer cancer prognosis and increased metastasis in many cancers [74]. ALDH are NAD(P)+-dependent enzymes responsible for catalysing the oxidation of aldehydes to carboxylic acids [75]. Due to their ability to detoxify toxic aldehydes protecting cancer cells from toxic ROS levels, ALDH was attributed to conferring chemotherapy drug-resistance, making ALDH inhibitors promising drugs against NPC [76].

#### 2.2.2. Methylation of Viral Genes

Due to the levels of histone and DNA methylation occurring in latent EBV cells, the availability of methyl donor groups is crucial in allowing these cells to remain in latency and consequently avoid immune detection by the host [77,78,79,80,81]. As described earlier, SAM is an important universal methyl donor that can be derived from the methionine and folate cycles. Under methionine restricted (MR) conditions in BL cells, the induction of latent highly immunogenic EBNA3, LMP1 and LMP2A of lower immunogenicity as well as lytic antigens including BZLF1 and BMRF1 [22,82] occurs. Transcriptomically, MR has also exhibited significant derepression of the previously listed latency associated proteins, along with 60 lytic genes associated with EBV [22].

The closely interlinked serine metabolic pathway also supports the generation of methionine. The inhibition of key enzymes serine hydroxymethyltransferase 1 and 2 (SHMT1 and 2) by the antagonist SHIN1 resulted in derepression of LMP1 and lytic BZLF1 [22]. Similarly to that of MR, transcriptomic data of SHMT inhibition also manifested in expression of the bulk of EBV genes—both latent and lytic [22]. As such, MR demonstrates the significance of modulating methionine availability to re-induce immunogenic EBV gene expression.

#### 2.2.3. Synthesis of Immunosuppressive Metabolites

Indoleamine 2,3-dioxygenase (IDO1) plays a catabolic role in the first step of the tryptophan–kynurenine–aryl hydrocarbon (Trp–Kyn–AhR) pathway, catalysing the reaction of L-tryptophan to N-formylkynurenine [83]. It has been found to be upregulated in a wide variety of cancers, including those that are EBV-related such as DLBCL. This upregulation in IDO1 interferes with the function and proliferation of CD8+ cytotoxic T lymphocytes that are important in the antitumor response by starving them of tryptophan, as well as suppressing NK cell activity by inhibiting the expression of activating receptor NK group 2, member D (NKG2D) receptor on the surface of NK cells [84,85]. The combination of IDO1 effects thus allows these cancer cells to escape immune targeting. IDO1 has been shown to be increased via the p38 MAP kinase (MAPK) and NF-κB pathways by mechanisms dependent on cytokines IL-6 and TNF-α, respectively, in EBV-infected human monocyte-derived macrophages (MDMs) instead of via the conventional IDO inducer—interferon-gamma (IFN-γ) [86]. Upon further investigation by Sawada et. al., Epstein–Barr virus-encoded RNA 1 (EBER1) delivered to EBV-infected cancer cells in the form of MDM-derived exosomes have been determined to upregulate IDO1 through IL-6 and TNF-α via retinoic acid-inducible gene-I (RIG-I) and the subsequent NF-κB and p38 MAPK pathway in EBV-infected cancer cells [83].

## 3. Metabolic Reprogramming in the Lytic Phase

### 3.1. Virus Production and Packaging

#### 3.1.1. DNA Replication

One interesting property of EBV is that it also carries genes to promote deoxyribonucleotide synthesis, e.g., thymidine kinase (designated as BXLF1) and large and small subunits of the viral ribonucleotide reductase (RNR) designated BORF2 and BaRF1, respectively [87,88]. Thymidine kinases catalyse the reaction converting deoxythymidine to deoxythymidine monophosphate, while RNRs function by catalysing the important reaction of converting nucleotides to deoxynucleotides de novo, making both enzymes crucial in enabling DNA replication and repair in any organism by generating precursors to deoxyribonucleotide triphosphates (dNTPs) to be used in DNA synthesis [89,90]. Both BORF2 and BaRF1 were reported to be expressed during the lytic stage in a variety of EBV-associated malignancies that include BL, LCLs, NKTLs and EBVaGC [9,91,92,93]. Intriguingly, Qiao et al. conclusively attributed the presence of EBV protein kinase instead of thymidine kinase to be the key factor activating both ganciclovir (GCV) and acyclovir (ACV)—key guanosine inhibitors targeting viral lytic replication [94].

#### 3.1.2. Protein Synthesis

The endoplasmic reticulum (ER) is a hub by which proteins are folded, including proteins of both host and viral origins when an infection occurs. Therefore, it is not surprising for the folding capacity of the ER to be overwhelmed by the introduction of large numbers of viral proteins required for the packaging of new virions arising in ER stress. EBV utilises early lytic protein, BMLF1, to significantly upregulate the expression of a key regulator in the unfolded protein response (UPR) pathway—GRP78 [95]. BMLF1 activates the protein activating transcription factor 6 (ATF6) via proteolytic cleavage, resulting in downstream expression of the *GRP78* (also known as *HSPA5* or *BiP*) gene to upregulate GRP78 protein [95]. It has been demonstrated that the knockdown of GRP78 significantly decreased virion production, emphasising the import of GRP78 in the assembly or release of EBV virions [95]. However, the means by which GRP78 achieves such an effect have not yet been ascertained, thus further studies should be warranted to allow for proper targeting of treatments.

#### 3.1.3. Lipid Synthesis

The fatty acid synthesis (FAS) pathway is especially important in the propagation of enveloped viruses as lipids are required for the formation of their membranes, aside from the necessary post-translational modification of palmitoylation required for proteins [96]. As an enveloped virus, EBV is known to upregulate the FASN in its lytic cycle. Activation of FAS in EBV has been attributed to the IE transcription factor BRLF1 via the stress-activated p38 MAPK pathway, similar to that in IDO1 [97]. Inhibition of FAS has been shown to suppress the expression of both BRLF1 and BZLF1, therefore underscoring the importance of FAS in lytic cells and highlighting FAS as a potential target against EBV lytic infection [98]. Nonetheless, the mechanisms by which FAS inhibition disrupts BRLF1 and BZLF1 expression have yet to be fully understood, hence creating a gap of understanding needing to be filled before this potential can be tapped on.

## 4. Clinical Trials

To highlight an increasing appreciation of metabolic pathways as viable therapeutic targets in EBV-driven cancers, we surveyed the International Clinical Trials Registry Platform (ICTRP) database for trials, both ongoing and completed, investigating the use of metabolic inhibitors or interventions for the treatment of EBV-driven malignancies (Table 1) [99].

Twenty years ago, nucleotide synthesis inhibitors became the mainstay of treating EBV cancers. Since then, the community has started looking towards using newer approaches such as immunotherapy (e.g., checkpoint inhibitors, CAR-T cells and autologous cell transfusions). This is evidenced by the increasing number of clinical trials combining these two modalities of treatment. However, caution is warranted in combining standard-of-care nucleotide inhibitors with immunotherapies; effector immune cells need to proliferate in order to perform their anti-tumour functions and inhibitors of nucleotide synthesis squarely prevent that from happening, resulting in minimised therapeutic efficacy. Hence, there is still scope for discovering new metabolic pathways that are cancer specific or which are at least more highly active in cancers than in bystander cells, for therapeutic exploitation.

## 5. Future Directions

Over the years, much ground has been covered in discovering key metabolic targets against EBV-related B cell malignancies—many of which we have touched upon in this review (Figure 1). Be that as it may, research into metabolic vulnerabilities in a non-B cell context—while becoming increasingly prevalent in recent years—still have huge gaps of understanding needing to be bridged. There is also an increasing appreciation for non-metabolic moonlighting roles of metabolic enzymes, e.g., regulation of RNA splicing [100]. It will be of interest to determine whether metabolic intervention can synergise with other treatment modalities to bring about augmented therapeutic efficacy in EBV-driven cancers and whether simple inhibition of enzymatic catalysis is sufficient to drive remission or that whole proteins have to be degraded in order to have the full therapeutic effects. The disparity in the depth of knowledge between EBV-related B cell and non-B cell cancers leaves much to be desired, especially with the majority of newly diagnosed EBV-related cancers being of non-B cell origins such as EBVaGC and NPC [101].

As alluded to earlier, there is still significant academic and industrial interest in developing metabolic inhibitors that are either more specific or more active in cancer cells. These include inhibitors targeting various metabolic pathways, including one-carbon inhibitors SHIN1 [102] and MTH-1479 [103] and de novo serine synthesis inhibitors CBR-5884 [102,104] and NCT-503 [105] as well as TCA inhibitors ivosidenib [106] and enasidenib [107]. However, many of these drugs are again targeting similar enzymes in healthy bystander cells and crucially, effector immune cells activated by immunotherapies. Hence, careful studies will be required to establish the therapeutic windows for achieving maximum tumour clearance with minimal side effects in bystander cells.

For lytic infection, a majority of antivirals directly target viral polymerases instead of working against host pathways that have been specifically modified by the viruses over the course of infection. The former has proven effective in most cases, though it should also be noted that nucleoside analogues mainly target late gene expression which is necessary for virion packaging. IE and E gene products as well as non-coding RNAs can also contribute to the pathogenesis of lytic EBV disease that includes chronic fatigue syndrome, oral hairy leukoplakia and chronic active EBV (CAEBV) [108,109,110]. It is unsurprising then that the latter approach is less attempted, as targeting these pathways may undeniably lead to a plethora of undesirable off-target effects, leaving a wellspring of possibilities in discovering therapeutic windows within viral-altered host pathways to achieve antiviral effects, though a careful balance between maximising antiviral capabilities and minimising toxicity must be struck. To date, there has only been a single study utilising a metabolic inhibitor against CAEBV (Table 1) [111]. However, it is promising to pair such a treatment in combination with acyclovir, a nucleotide inhibitor. There are a limited number of patients and types of cell lines that allow for deeper mechanistic studies into lytic EBV diseases. Nonetheless, given our current understanding of the immense metabolic demands of lytic replication, there is untapped potential in using metabolic inhibitors on top of the standard-of-care nucleotide inhibitors.

## Figures and Tables

**Figure 1 cancers-15-03412-f001:**
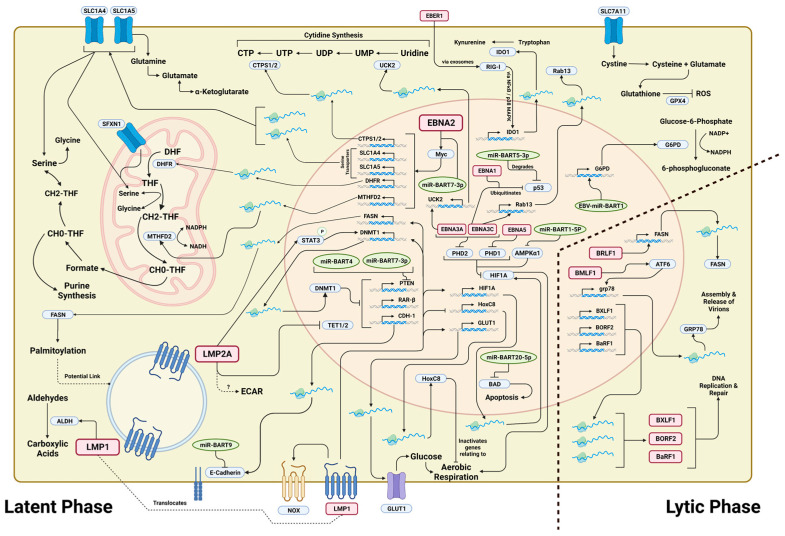
Schematic of metabolic pathways altered by EBV in both latent and lytic phases. Viral proteins are boxed in red, proteins in blue and miRNAs are circled in green. Image was created with BioRender.com.

**Table 1 cancers-15-03412-t001:** Compilation of past and existing clinical trials covering metabolic pathways.

Metabolic Pathways Implicated	Specific Factor/Molecule(s) Targeted by Intervention	Inhibitor	Trial ID
Amino acid metabolism	Extracellular asparagine and glutamine	Asparaginase	JPRN-UMIN000003498
Folate metabolism	Dihydrofolate reductase (DHFR)	Methotrexate	NCT00822432 NCT01964755
Glucose metabolism	Glycolytic enzymes	N.A. *	ChiCTR-ROC-15006026 NCT02481089
	Monocarboxylate transporter 1 (MCT1; also known as SLC16A1)	AZD3965	NCT01791595
Iron metabolism	Free iron	Deferasirox	NCT01159067 NCT01273766
	Iron transporters	N.A. *	ChiCTR2100042554
Nucleotide metabolism	Dihydropyrimidine dehydrogenase (DPD)	Eniluracil	NCT00264472
		Gimeracil	ChiCTR1800015670
	Ribonucleotide reductase (RNR)	Gemcitabine	ChiCTR-ONC-12002613 ChiCTR1900022288 ChiCTR1900027112 ChiCTR2100041804 CTRI/2020/10/028269 EUCTR2010-022444-20-NL KCT0003189 KCT0006096 NCT00060112 NCT00072514 NCT00370890 NCT00436800 NCT00690872 NCT00697905 NCT01309633 NCT01417390 NCT01528618 NCT01596868 NCT01854203 NCT02016417 NCT02460887 NCT02578641 NCT02789189 NCT02878889 NCT03639467 NCT03707509 NCT04405622 NCT04458909 NCT04517214 NCT04522050 NCT04833257 NCT04890522 NCT04898374 NCT05062005 NCT05294172 NCT05340270 NCT05484375 NCT05576272 NTR2740
		Hydroxyurea	NCT00180973 NCT01964755
	Thymidylate synthase	5-fluorouracil	ChiCTR-TRC-13003378
		Capecitabine	NCT04072107
		Tegafur	ChiCTR1800015670
Nutrient Signalling via mTOR	FK506-binding protein 12 (FKBP12)	RAD001	NCT01341834

* N.A.—No drug or treatment has been ascribed to the trials as genes are being studied.

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
