# Peer review of "Targeting Metabolic Vulnerabilities in Epstein–Barr Virus-Driven Proliferative Diseases"

_cancers, 2023, doi:10.3390/cancers15133412_

Round 1
Reviewer 1 Report
This comprehensive review of EBV-related mechanisms of oncogenesis and tumor progression focuses on cell metabolism. Potential for therapy targeting metabolic biochemical pathways is discussed. Figure 1 is quite comprehensive, and it is worth considering if, in addition, a few more figures could be created to accompany selected sections of the text to illustrate the pathways highlighted in that section.
Suggestions to improve:
1. Line 39, Include NPC in the introductory list of EBV-associated diseases.
2. Lines 311 and 348, in these sections, distinguish histone methylation from DNA methylation.
3. Italicize gene symbols but not protein symbols, e.g. PTEN in Line 323, to clarify when genes vs proteins are meant.
Minor rewording suggestions:
4. Consider rewording ‘intra-organismal spread through metastatic invasion’ to ‘spread through metastatic invasion’.
5. Change Hodgkin’s to Hodgkin.
6. Line 81, change ‘trials’ to ‘clinical trials’
7. Consider rewording ‘metabolic nodes’ to ‘metabolic factors’
8. Line 156, 158, the word ‘express’ needs to be changed.
9. Line 186, reword to clarify which protein has isoforms
1. Line 217, consider changing ‘lymphoid development’ to ‘tumor burden’
1. Lines 263 and 311 have the same section numbers.
1. Line 323, typo
1. Line 445, The legend says red, but the miRs appear to be in green.
Very dense text.
Author Response
Comment: Line 39, Include NPC in the introductory list of EBV-associated diseases.
We thank the reviewer for the comment. We made the change as suggested.
Comment: Lines 311 and 348, in these sections, distinguish histone methylation from DNA methylation.
We thank the reviewer for the comment. We made the change as suggested.
Comment: Italicize gene symbols but not protein symbols, e.g. PTEN in Line 323, to clarify when genes vs proteins are meant.
We thank the reviewer for the comment. We made the change as suggested in lines 104, 147-149, 327 and 413.
Comment: Consider rewording ‘intra-organismal spread through metastatic invasion’ to ‘spread through metastatic invasion’.
We thank the reviewer for the comment. We made the change as suggested.
Comment: Change Hodgkin’s to Hodgkin.
We thank the reviewer for the comment. We made the change as suggested.
Comment: Line 81, change ‘trials’ to ‘clinical trials’
We thank the reviewer for the comment. We made the change as suggested.
Comment: Consider rewording ‘metabolic nodes’ to ‘metabolic factors’
We thank the reviewer for the comment. We made the change as suggested.
Comment: Line 156, 158, the word ‘express’ needs to be changed.
We thank the reviewer for the comment. We have clarified the direction of expression change.
Comment: Line 186, reword to clarify which protein has isoforms
We thank the reviewer for the comment. We clarified this by stating that GLUT1 is more affected than other glucose transporter family members.
Comment: Line 217, consider changing ‘lymphoid development’ to ‘tumor burden’
We thank the reviewer for the comment. We made the change as suggested.
Comment: Lines 263 and 311 have the same section numbers.
We thank the reviewer for the comment. We made the change as suggested.
Comment: Line 323, typo
We thank the reviewer for the comment. We have rectified the typographical error.
Comment: Line 445, The legend says red, but the miRs appear to be in green.
We thank the reviewer for the comment. We have rectified the error.
Reviewer 2 Report
In the review “Targeting Metabolic Vulnerabilities n Epstein-Barr Virus-driven Malignancies”, the authors summaries the contributions of metabolic pathway induced by EBV and discussed the potential treatment targeting metabolic vulnerabilities in EBV-associated diseases.
I would suggest to discuss the role of he enzyme IDO1 in the macrophage polarization and in the pro-tumor response of EBV-associated malignancies and add some reference in the section of “Molecular Metabolic mechanism driving immune evasion and metastasis”
The manuscript is generally well-written
Author Response
Comment: I would suggest to discuss the role of the enzyme IDO1 in the macrophage polarization and in the pro-tumor response of EBV-associated malignancies…
We thank the reviewer for the comment. We have attempted to do a literature search on studies linking IDO1 and macrophage polarization to EBV infection, however, we were unable to find studies that combined and addressed these concepts as interlinked. Hence, we regret that no change has been made in response to this particular comment.
Comment: …and add some reference in the section of “Molecular Metabolic mechanism driving immune evasion and metastasis”
We thank the reviewer for the comment. We now have 15 references in that section.